# SERCNN: STACKED EMBEDDING RECURRENT CONVOLUTIONAL NEURAL NETWORK IN DEPRESSION DETECTION ON TWITTER

## ABSTRACT

Conventional approach of self-reporting-based screening for depression is not scalable, expensive, and requires one to be fully aware of their mental health. Motivated by previous studies that demonstrated great potentials for using social media posts to monitor and predict one's mental health status, this study utilizes natural language processing and machine learning techniques on social media data to predict one's risk of depression. Most existing works utilize handcrafted features, and the adoption of deep learning in this domain is still lacking. Social media texts are often unstructured, ill-formed, and contain typos, making handcrafted features and conventional feature extraction methods inefficient. Moreover, prediction models built on these features often require a high number of posts per individual for accurate predictions. Therefore, this study proposes a Stacked Embedding Recurrent Convolutional Neural Network (SERCNN) for a more optimized prediction that has a better trade off between the number of posts and accuracy. Feature vectors of two widely available pretrained embeddings trained on two distinct datasets are stacked, forming a meta-embedding vector that has a more robust and richer representation for any given word. We adapt Lai et al. (2015) RCNN approach that incorporates both the embedding vector and context learned from the neural network to form the final user representation before performing classification. We conducted our experiments on the Shen et al. (2017) depression Twitter dataset, the largest ground truth dataset used in this domain. Using SERCNN, our proposed model achieved a prediction accuracy of 78% when using only ten posts from each user, and the accuracy increases to 90% with F1-measure of 0.89 when five hundred posts are analyzed.

## 1 INTRODUCTION

Depression is a serious yet common mental disorder that affects more than 264 million people worldwide. The number is projected to grow amid the war against the pandemic of COVID-19. Unlike the usual mood fluctuations and emotional responses, the long-lasting sadness, emptiness, or irritation in one's day-to-day life, accompanied by somatic and cognitive changes that heavily disrupt an individual's capacity to function normally. Hence, depression is often associated with suicide at its worst. In the latest Word Health Statistics 2021 (World Health Organization, 2021), 28% increases the suicide rate in the United States in this period, which may result from the increasing pressure on the loss of loved ones and incomes, city lockdowns, social distancing, and the reduction of human interactions.

Existing screening methods, such as psychometric self-reporting questionnaires and clinical interviews, are expensive, not scalable, not reachable to many, and have limitations. For example, the Diagnostic and Statistical Manual of Mental Disorders (DSM-5) (American Psychiatric Association et al., 2013) requires patients to be involved and disclose truthful information during clinical interviews. Besides that, people who suffer from depression tend to be unaware of their condition because there is no specific feedback from the body like common physical injuries do. To get diagnosed with depression, one must first consult a doctor or mental health professional. Up to date, global provision and service for identifying, supporting, and treating mental health issues are insufficient despite the disruption of essential health services, community mistrust, and fears of COVID-19

infections. The lack of mental health awareness in the Southeast Asia Region is another contributing factor in the sky-high figure of depression.

Social media has become part and parcel of everyday life. It has become a valuable source of information that led to a new research direction for detecting depression. Here, social media data can serve as a lifelog that records users' activities in the text, image, audio, and video. The enormous amount of time series first-person narrative posts can provide insights about one's thoughts, feelings, behavior, or mood for some time, allowing an unintrusive way to study patients' conditions before an actual clinical diagnosis is made.

Motivated by previous studies that demonstrated great potentials for using social media posts to monitor and predict one's mental health status, this paper utilizes natural language processing and machine learning techniques on social media data to predict one's risk of depression. We propose Stacked Embedding Recurrent Neural Network (SERCNN), which can effectively learn the representation from dirty and unstructured social media text without the need to pursue multimodal learning. Instead of using the recent transformer model, we suggest a meta-embedding approach for our feature extraction. We utilize two widely available yet low dimensional pretrained GloVe embeddings trained on different domain datasets to provide a more robust and accurate representation of the dirty and unstructured social media text that often contains typos. Our SERCNN can achieve a state-of-the-art performance without using all the social media posts in the user posting history, providing a perspective on the number of posts required to make a reliable prediction. It is worth pointing out that there is still room in social media text that is worth exploring.

The contributions of this paper can be summarized as below.

1. SERCNN can retrieve a more robust and richer representation of the social media text by stacking multiple low dimensional pretrained embeddings.

2. Provides insights on how the different number of posts influence the prediction model's accuracy.

## 2 RELATED WORK

In this section, we provide an overview of the current research landscape and how previous works convert information extracted from social media posts into more representative features that can be used to identify depressed individuals.

**Depression detection on social media** Evidence from various published works has shown the viability of using social media data to predict depression and other mental disorders. The seminal work by Holleran (2010) has inspired natural language processing (NLP) researchers to identify potential markers of mental disorder from social media posts. In contrast to the 14 days of observation stated in DSM-5, Hu et al. (2015) and Tsugawa et al. (2015). empirically suggested that features extracted over two months are sufficiently be used to identify one's depression condition. Coppersmith et al. have presented a. novel method to collect ground truth on social media related to depression and PTSD. The dataset is then being used for the shared task of The Workshop on Computational Linguistics and Clinical Psychology (CLPsych) (Coppersmith et al., 2015). A similar workshop, CLEF eRisk (Losada et al., 2017), focuses on the early depression symptoms discoveries on social media. Workshop tasks such as depression detection (CLPsych 2015 (Coppersmith et al., 2015)) and early detection of signs of depression (CLEF eRisk 2017 (Losada et al., 2017)) have generated significant numbers of novel approaches in identifying depression on social media.

**Feature representation for depression classification** Modeling feature representations is a crucial task in machine learning; features that are not discriminative and representative will result in poor and faulty model performance. Hence, earlier research works are mainly focused on feature extraction techniques. Choudhury et al. (2013); Tsugawa et al. (2015) have found that depressed users tend to be emotional. Wang et al. (2013) found that using sentiment analysis in depression detection can achieve about 80% accuracy. Tsugawa et al. (2015); Resnik et al. (2015) extracted topics distribution with Latent Dirichlet Allocation (LDA) (Blei et al., 2003) to differentiate depressed individuals from the healthy controls. Researchers also extracted features based on criteria stated in the industry-standard - Diagnostic and Statistical Manual of Mental Disorders (DSM), such as the insomnia index derived from the user posting time. Linguistic Inquiry and Word Count (LIWC) (Tausczik & Pen-

nebaker, 2010) is a widely used word matching-based feature extraction tool that builds on top of Pennebaker et al.'s findings (Rude et al., 2004; Gortner et al., 2006) decades ago. (Choudhury et al., 2013; Shen et al., 2017) show that depressed users tend to have high self-attentional focus, increased medicinal concerns, and increased expression of religious thoughts. These findings are aligned with the Rude et al. (2004) work where depressive indicators can be found on the human-generated content.

However, LIWC and word matching-based sentiment analysis approaches are often ineffective. The nature of social media text being dirty and unstructured often required extensive data cleaning and preprocessing for the tools to act as intended. Recent works (Gui et al., 2019a;b; Rao et al., 2020) have shown that using the word embedding and deep learning models are effective than the hand-crafted features with minimal effort of text preprocessing. We observed that most works that employed deep learning models tend to use hierarchical document modeling to generate their input representation. This raises the question of whether depression classification using social media data actually takes advantage of the hierarchical document modeling.

# 3 STACKED EMBEDDING RECURRENT CONVOLUTIONAL NEURAL NETWORK (SERCNN)

**Overview** Our goal is to improve the performance of depression classification on social media by learning a much robust user representation. Given that our dataset, $d$, consists of $N_1$ number of social media users, $u$, where each user's $N_2$ number of social media posts, $p$, within a month were collected, and each post has $N_3$ words, $w$, we denote the dataset as $d = \{u_1, ..., u_{N_1}\}$, social media user as $u = \{p_1, ..., p_{N_2}\}$, and the post as $p = \{w_1, ..., w_{N_3}\}$.

We propose Stacked Embedding Recurrent Convolutional Neural Network (SERCNN) which is made up of a Stacked Embedding (SE) and Lai et al. (2015) Recurrent Convolutional Neural Network (RCNN). The overall architecture of SERCNN is simple, consisting a SE layer, single directional LSTM, a max-pooling layer and an output layer which is a fully connected layer with sigmoid function, as visualized in Figure 1.

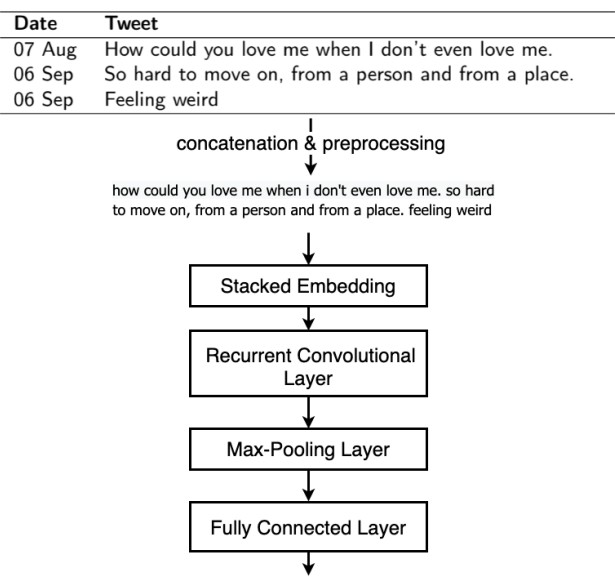

Figure 1: The overall architecture of the proposed SERCNN.

### 3.1 FEATURE EXTRACTION WITH STACKED EMBEDDING

Unlike previous works, we model our user representation by firstly concatenating $N_2$ social media posts in chronological order. For a random user $u_i$, the concatenated text representation can be formulated into:

$$u_i = \{p_1 + ... + p_{N_2}\} \tag{1}$$

$$u_i = \{w_1^1 + ... + w_{N_3}^1 + ... + w_{N_3}^{N_2}\} \tag{2}$$

This concatenated post can be viewed as a single "monthly diary", journal or large document that characterizes the user.

Then, we extract the distributed text representation of each word with the SE, which is an ensemble embedding, commonly known as the Meta-embedding technique. The concept of Meta-embedding was first introduced by Yin & Schütze (2016) to utilize and learn the meta of existing well-trained pretrained embeddings and extend the vocabulary. Since different pretrained embeddings were trained on different datasets, each embedding can now complement each other allowing an improved vocabulary coverage and reducing out-of-vocabulary words. As the name suggests, SE is formed by stacking the collection of dense vectors (pretrained weights) $\boldsymbol{E} = \{\boldsymbol{E}_1, ..., \boldsymbol{E}_{N_4}\}$ extracted from $N_4$ number of pretrained embeddings included:

$$\boldsymbol{E}_{SE} = \{\boldsymbol{E}_1 + ... + \boldsymbol{E}_{N_4}\} \tag{3}$$

where the embedding vector, $\boldsymbol{x}$, of a given word, $w$ can be obtained via:

$$\boldsymbol{x} = \boldsymbol{E}_{SE}(w) \tag{4}$$

The vocabulary of SE, $V_{SE}$, is now considered as the vocabulary union of the $N_4$ pretrained embeddings, resulting in a more extensive vocabulary than a single embedding:

$$V_{SE} = \bigcup_{n=1}^{N_4} V_n \tag{5}$$

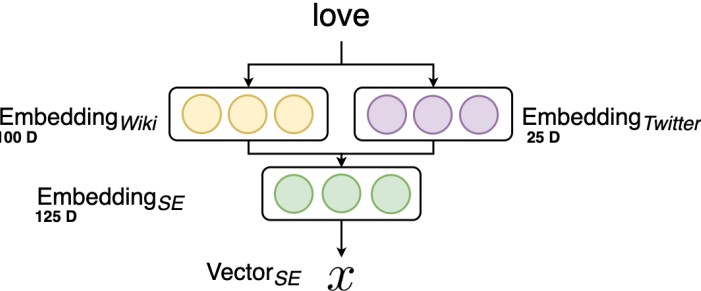

Figure 2: Illustration of st

In this study, our SE is the simple stacked vector made up of two pretrained GloVe embeddings (Pennington et al., 2014) trained on Twitter and Wikipedia 2014 + Gigaword 5 datasets, respectively:

1. ***GloVe Twitter (25 dimensions)*** trained using global word co-occurrences information by Pennington et al. (2014) under an uncased setting, using 2 billion tweets, with 27 billion tokens. The resulting vector consists of 1.2 million vocabularies learned from the corpus.

2. ***GloVe Wikipedia 2014 + Gigaword 5 (100 dimensions)*** (Pennington et al., 2014), similar to the GloVe Twitter embedding, this embedding is trained using global word co-occurrences information but using a different corpus, which is the combination of Wikipedia 2014 and Gigaword 5 datasets. There are approximately 400 thousand words in the vocabulary.

## 3.2 Representation Learning and Depression Classification with RCNN

Recurrent neural network (RNN) is capable of capturing contextual information over a long sequence. However, the RNN model favors the later words than words in the earlier sequence. In depression detection on social media, where posts are collected over a time interval, we are interested in identifying words throughout that period, rather than just words that occurred later. Lai et al. (2015) Recurrent Convolutional Neural Network (RCNN) overcomes the limitation of the existing RNN model by incorporating a max-pooling layer to extract key features from both the embedding features and the context learned. The max-pooling layer reconsiders embedding features instead of just the context learned in the conventional RNN-based setting, selecting the important features for the classification task.

Figure 3: Recurrent Convolutional Neural Network (RCNN) using single Forward LSTM

The overall architecture of our RCNN model is visualized in Figure 3. Instead of using bidirectional Long Short Term Memory (LSTM) as described in Lai et al. (2015), we use a generic single Forward LSTM to learn the context, $c$, from the embedding vector, $v$. With $j$ refers to the $j^{th}$ social media post and $k$ refers to the $k^{th}$ number of words in the post, we can formulate the context for a given word:

$$\left(\boldsymbol{c}(w_k^j), \boldsymbol{h}(w_k^j)\right) = \text{LSTM}\left(\boldsymbol{c}(w_{k-1}^j), \boldsymbol{h}(w_{k-1}^j), \boldsymbol{v}(w_k^j)\right) \tag{6}$$

where $|\boldsymbol{c}| \in \mathbb{R}$ and $\boldsymbol{h}$ is the output vector.

The output of the LSTM is then concatenated with the embedding features, forming an extended context vector:

$$\boldsymbol{y}_k^j = [\boldsymbol{c}_{(}w_k^j); \boldsymbol{x}_k^j] \tag{7}$$

A max-pooling layer is applied after the RCNN representation of the user is computed:

$$\tilde{\boldsymbol{y}} = \max_{l=1}^{L} \boldsymbol{y}_l \tag{8}$$

, where the total number of words, $L$, can be calculated by multiplying the total number of social media posts, $N_2$, by the number of words in each post, $N_3$.

A fully connected layer is then used to discriminate the max-pooled context with a sigmoid function, presenting the classification output, $\hat{\boldsymbol{y}}$, as probabilities:

$$\hat{\boldsymbol{y}}_i = p\left(\tilde{\boldsymbol{y}} \mid u_i\right) = \frac{1}{1 + exp^{-(\boldsymbol{W}_c u_i + b_c)}} \in [0, 1] \tag{9}$$

## 4 Experiments

**Benchmark dataset and data preprocessing** We employed Shen et al. (2017) Depression Twitter dataset, which is by far the largest benchmark dataset available at the time of submission. The ground truth is collected based on the user's self-disclosure of their depression diagnosis, which

satisfied the strict pattern "(I'm/ I was/ I am/ I've been) diagnosed depression". Users who never mention the character string "depress" in their posting history were included as the healthy controls.

To ensure a fair comparison against previous papers, we do not expand the dataset and use the data included in the dataset only. For data preprocessing, we performed:

1. ground truth (anchor tweet) removal, to avoid the model being biased towards the string character "depress".

2. retweet removal, to focus on the genuine content generated by the user.

3. url removal.

4. non-English healthy controls removal, to reduce noise in the training data.

5. lowercasing, as we are using the pretrained GloVe uncased embeddings.

6. splitting the symbol '@' from the user mentioned, to better characterize a subject entity, instead of an out-of-vocabulary word. For example, "@ICLR2022" becomes "@" and "ICLR2022

After performing data preprocessing, we realized that there is one of the ground truths consists only of retweets. Therefore, we removed that user and randomly select one from the non-depression dataset to form a balanced dataset. The statistic of the dataset is as shown below:

Table 1: Dataset statistics

| Dataset | | Number of users | Number of posts | Average posts per user |
|---------|--|-----------------|-----------------|------------------------|
| D1 | Depression | 1,401 | 231,494 | 165 |
| D2 | Non-depression | 1,401 | 1,119,080 | 799 |
| Total | | 2,802 | 1,350,574 | 482 |

**Experiment settings** We trained our model using PyTorch 1.6 and torchtext 0.7 libraries with CUDA on an Nvidia P5000. All models have trained with Adam (Kingma & Ba, 2015) optimizer and cross entropy function as the loss function. We perform grid search to find the optimized hyperparameters for both SERCNN and baselines; the hyperparameters include learning rate, hidden layer dimension, number of vocabulary, dropout, training epochs, early stopping criteria, and batch size are reported in Table 2.

Table 2: Hyperparameter settings

| Hyperparameter | Search range | Optimized |
|----------------|--------------|-----------|
| learning rate | {0.0005, 0.001, 0.002, 0.01} | 0.001 |
| hidden layer dimension | {100, 200, ... , 1000} | 500 |
| number of vocabulary | - | 100,000 |
| dropout | - | 0.5 |
| training epochs | - | 100 |
| early stopping criteria | - | 20 |
| batch size | - | 1 |

For the choice of embeddings for SE, we select GloVe Twitter with 25 dimensions and GloVe Wikipedia with 100 dimensions, empirically, as described in Section 3.1.

## 5 RESULTS AND ANALYSIS

We benchmarked our SERCNN with the best models reported by Shen et al. (2017); Gui et al. (2019a;b) that are trained on the same dataset. Specifically, we compare the Shen et al. (2017) Multimodal Dictionary Learning (MDL) and Multiple Social Networking Learning (MSNL), which is trained with handcrafted features, Gui et al. (2019a) CNN and LSTM with Policy Gradient Agent (PGA), and Gui et al. (2019b) Gated Recurrent Unit (GRU) + VGG-Net + Cooperative Misoperation Multi-Agent (COMMA) policy gradients. In addition, we trained an LSTM model that used the concatenation representation proposed in Section 3 as well as a Yang et al. (2016) Hierarchical

Table 3: Performance comparison against baselines

| Model | Training data | Acc | Pre | Rec | F1 |
|---|---|---|---|---|---|
| Shen et al. (2017) MSNL | Handcrafted | 0.818 | 0.818 | 0.818 | 0.818 |
| Shen et al. (2017) MDL | features | 0.848 | 0.848 | 0.85 | 0.849 |
| Gui et al. (2019a) CNN | Text | 0.843 | 0.843 | 0.843 | 0.844 |
| Gui et al. (2019a) CNN + PGA | (Hierarchical) | 0.871 | 0.871 | 0.871 | 0.871 |
| Gui et al. (2019a) LSTM | | 0.828 | 0.830 | 0.828 | 0.828 |
| Gui et al. (2019a) LSTM + PGA | | 0.870 | 0.872 | 0.870 | 0.871 |
| Gui et al. (2019b) GRU + VGG + COMMA | Text + image | 0.900 | 0.900 | 0.901 | 0.900 |
| 1EHAN | Text | 0.906 | 0.920 | 0.906 | 0.913 |
| SEHAN | (Hierarchical) | 0.931 | 0.978 | 0.931 | 0.954 |
| 1ELSTM | Text | 0.900 | 0.921 | 0.900 | 0.910 |
| SELSTM | (Concatenated) | 0.920 | 0.936 | 0.920 | 0.928 |
| 1ERCNN | | 0.893 | 1.0 | 0.895 | 0.944 |
| SERCNN (10 posts) | | 0.784 | 0.809 | 0.764 | 0.770 |
| SERCNN (500 posts) | | 0.899 | 0.879 | 0.915 | 0.893 |
| SERCNN (all posts) | | 0.914 | 1.0 | 0.905 | 0.949 |

Attention Network (HAN) that used the hierarchical document modeling approach in both single embedding (prefix of 1E) and Stacked Embedding (prefix of SE) settings. 1E models are trained with pretrained GloVe Wikipedia 300 dimensions. Performances of the models are evaluated with accuracy, macro-averaged precision, macro-averaged recall, and a macro-averaged F1-measure.

From Table 3, we can observe that the SERCNN and the baselines we trained outperformed previous works Shen et al. (2017); Gui et al. (2019a;b). This significant performance gap is due to these aspects we observed:

- **Posts included in the study** We exclude all non-genuine posts (retweets) created by other users because these posts do not reflect the user writing style. In a similar action, Gui et al. (2019a) implemented a policy gradient model to selectively exclude irrelevant social media posts in generating their user representation and increase F1-measure by 4.3% for the LSTM and 2.7% for the CNN models. However, having a policy gradient model to control the input might yield bias and leave out some important information not previously identified. Therefore, we suggest including only the genuine user-generated content by the users is sufficient to determine one's depression condition.

- **Richer representation** The combination of SE and RCNN aims to enrich existing representations. Our reported results show that stacked embedding yields a more robust representation than a high dimensional pretrained embedding, where all models implemented (with the prefix SE-) have improved performance, with HAN gains a 4.1 % increment, and LSTM gains 1.8 % in F1-measure. Even though there is a minimal improvement using SE over 1E on RCNN, 1ERCNN achieved a better F1-measure performance than all other models trained using a single embedding and a comparable performance with SEHAN and SELSTM. This empirical result suggested that the LSTM might neglect or miss some crucial information, as the main difference between the RCNN and LSTM is the additional step in concatenating the word embedding features with the LSTM context. The reintroduction of embedding features allows the model to have a second chance to understand the context better.

- **Concatenated post** A good feature representation is vital to the performance of any classification model. Representing our input as a single concatenated text sequence allows 1ELSTM and SELSTM to have a competitive performance over the 1EHAN and SEHAN, which are more sophisticated models. The hierarchical modeling approach learns the context of each social media posts independently first before forming the global context. As stated in the previous paragraph, the context learned is not necessarily the most discriminative. Consequently, the underlying relationships across different posts might not be captured correctly. Earlier works (Shen et al., 2017) have suggested that first-person pronouns are solid features, but they are often treated as stop words and removed from the corpus.

In contrast, presenting the input as a single text sequence allows the model to learn and exploit words occur in posts in different period.

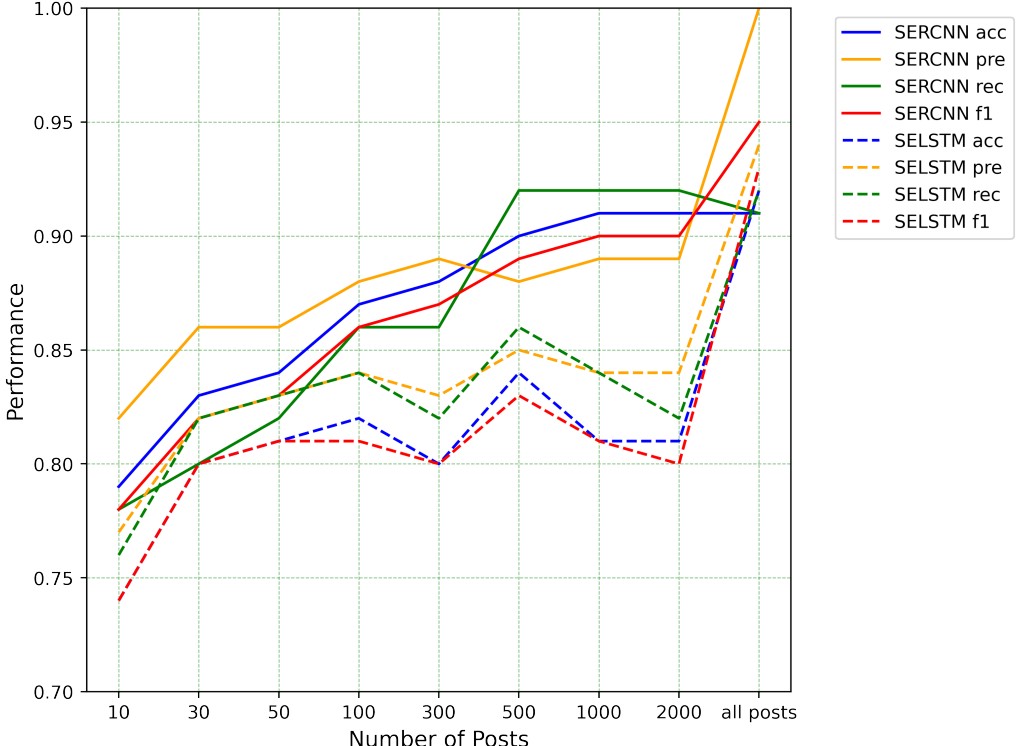

Figure 4: Performance of SERCNN and SELSTM trained on different number of posts

**Finding optimal number of posts** The performance of our proposed SERCNN is almost identical to its single embedding counterpart, but these models' performances are topped in the Table 3. This observation suggested that RCNN architecture can capture significant embedding features that are neglected by LSTM and relate these features with the context learned by LSTM in discriminating depression users and healthy controls. From the identical results, we can deduce that the performance reaches the state of plateau when we include all social media posts collected within a month.

We have reconstructed the dataset into eight variants, which these datasets are made up of the most recent $\{10, 30, 50, 100, 300, 500, 1000, 2000\}$ posts, respectively. From the result obtained, we found that using as minimal as ten posts for each user, SERCNN can achieve high accuracy and F1-measure of 0.793 and 0.781. This result is significant because the observation of just ten posts in a month is very appealing compared to the two-month observation window suggested by both Hu et al. (2015) and Tsugawa et al. (2015). Through experiments, the optimal number of maximum posts is found to be 500; without compromising the performance, SERCNN achieves 89.9% accuracy and an F1-measure of 0.89. The optimal number obtained is closed to the average posts per user included in this study at 482 posts.

Table 4: Complexity analysis of SERCNN and baselines

| Model | Sequential operation | |
|---|---|---|
| LSTM | | $\mathcal{O}(N_2 N_3)$ |
| RCNN | | $\mathcal{O}(N_2 N_3)$ |
| Hierarchical LSTM | Word level: | $\mathcal{O}(N_2 N_3)$ |
| | Post level: | $\mathcal{O}(N_2)$ |

**Complexity Analysis of SERCNN** Table 4 reports the time complexity analysis of SERCNN with LSTM trained with concatenated posts and hierarchical LSTM (hierarchical document modeling), all LSTM trained are single Forward LSTM models. We use the notation we defined in Section 3, where $N_2$ refers to the number of posts and $N_3$ refers to the number of words. Since all three LSTM models run sequentially from the first word to the last word, the base sequential operation can be measured evaluated into $\mathcal{O}(N_2N_3)$. However, for the hierarchical LSTM, it incurs a higher complexity of $\mathcal{O}((N_2)^2N_3)$ as it introduced an additional hierarchical structure at post level ($\mathcal{O}(N_2)$) to learn the global context from the post context learned.

## 6    Conclusion and Future Work

Based on this finding, a cost-effective and scalable early intervention solution can be developed to improve global well-being, as SERCNN does not require massive computing power and storage. Concatenating social media posts into a single diary (document) allows the RNN-based model to exploit the relationship of words in a different time frame, resulting in a generalized global context for a user. Stacked Embedding allows various pretrained embeddings from different benchmark datasets to complement each other and reduce the out-of-vocabulary words for much robust classification. SERCNN exploits the rich representation from Stacked Embedding, and the context of the user representation learned, allowing it to outperform previous works. Besides that, our experiment demonstrates that SERCNN can achieve about 78% accuracy using 10 posts only and 90% accuracy using only 500 posts, without the need to utilized the whole dataset.

For our future work, we would like to extend our work with interpretability and explainability mechanisms. Providing reasons for the classification decisions is especially crucial with the recent enforcement of the European GDPR Law, where people have the right to explanation.

### Reproducibility Statement

The code will be made available once the findings are published.

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
