# OpenReview forum: "SERCNN: Stacked Embedding Recurrent Convolutional Neural Network in Depression Detection on Twitter"
_ICLR.cc/2022/Conference — ICLR 2022 Submitted_

### Official Review · Reviewer_4NmL · 2021-10-30

**Correctness:** 3
**Technical Novelty And Significance:** 1
**Empirical Novelty And Significance:** 1
**Recommendation:** 3
**Confidence:** 5

**Main Review:**

+Ves
+ The depression detection application is valuable, and the method achieves good performance.
+ The whole structure is nice.

-Concerns

-The key concern about the paper is the lack of novelty. The RCNN method is classic, and the word embedding concatenation is also very common. So, the novelty of this paper is not enough.

-The motivation is not clear. The reason why the authors use RCNN and what problem the authors want to solve should be explained, rather than only for getting better results.

-The paper says “Instead of using the recent transformer model”, Why not use transformer-based methods, like BERT? The authors should explain it and add comparative experiments.

-In the experimental results, why does the EHAN achieve the best results and not the proposed ERCNN? In addition, the paper says that the ERCNN can achieve best performance when using a few posts, but there are no comparative results that contain results of other methods with 10 posts or 100 posts.

-The authors should add ablation studies to show the effectiveness of stack embedding.

-The related works are not sufficient and many latest literature are missing.

Minor comments:

* Different evaluation metrics should not be put in one figure (Fig. 4).

* Some references should be added for background.


**Summary Of The Paper:**

This paper proposes a stacked embedding recurrent neural network, named SERCNN, to detect depression from Twitter. First, the authors use stacked meta-embedding to gain the stacked word information. Then, the RCNN structure is utilized to capture contextual features. The experimental results show the effectiveness of the proposed model.

**Summary Of The Review:**

The novelty is not enough, and many experiments are missing.

---

### Official Review · Reviewer_pXvC · 2021-10-30

**Correctness:** 2
**Technical Novelty And Significance:** 1
**Empirical Novelty And Significance:** 2
**Recommendation:** 3
**Confidence:** 5

**Main Review:**

### Strength

- (S1) SERCNN shows solid improvements over existing solutions on the Twitter depression detection dataset.


### Weaknesses

- (W1) Novelty and technical contributions are not significant.
- (W2) The paper lacks comparisons against pre-trained Transformer-based language models.
- (W3) No insights about depression are provided.


### Major comments

(W1)
SERCNN is a straightforward combination of existing techniques. Essentially, the paper combines 2x GloVe word embeddings into a single vector (without using any learning mechanism) for each input word and then uses the concatenated vector as input to an RCNN model.


(W2)

The author(s) mention the reason why they prefer RCNN over RNN is to capture context information over a long sequence, which can be addressed by using the self-attention mechanism.

The paper does not compare with any Transformer-based models, especially pre-trained Transfomer-based language models (e.g., BERT or more recent models,) which are considered more label efficient than conventional word embedding models (e.g., GloVe) + RNNs (technically, this paper uses RCNN.)

In fact, pre-trained language models were already used in the context of emotional detection from text. Thus, it is not natural to disregard the technique for depression detection. For example,

- [1] Yen-Hao Huang, Ssu-Rui Lee, Mau-Yun Ma, Yi-Hsin Chen, Ya-Wen Yu, Yi-Shin Chen, EmotionX-IDEA: Emotion BERT – an Affectional Model for Conversation, SocialNLP 2019. (https://arxiv.org/abs/1908.06264)
- [2] Kisu Yang, Dongyub Lee, Taesun Whang, Seolhwa Lee, Heuiseok Lim, EmotionX-KU: BERT-Max based Contextual Emotion Classifier, SocialNLP 2019. (https://arxiv.org/abs/1906.11565)


(W3)

I would expect to see new findings of depression detection on Twitter. The paper simply presents a technique (which has issues with respect to novelty and technical significance, as commented above) and claims the contribution by showing the numbers. This point may not be the main scope of ICLR (i.e., which may put more emphasis on technical contributions,) but I believe it is very important for application-oriented papers.


### Minor comments

- (Presentation style) Figure 2 is misleading. The figure looks like a linear layer is applied on top of the concatenated vector, which is not (i.e., the concatenation.)


**Summary Of The Paper:**

This paper develops a SERCNN, which consists of stacked embeddings and Recurrent CNN (RCNN), for depression detection from Twitter text. The key idea of stacked embeddings is to concatenate two embeddings, based on two different word embeddings models (pre-trained on Twitter and Wikipedia copora), into a single embedding vector, which is fed into RCNN. Experimental results on the Twitter Depression dataset show reasonable performance when trained on 10 posts from each user, which was further improved when trained on more data.


**Summary Of The Review:**

Although the paper tackles an important problem, it does not have a sufficient level of novelty and technical contribution. SERCNN is a straightfoward combination of existing methods. SERCNN relies on pre-trained word embedding models. Pre-trained Transformer-based language models should be compared.

---

### Official Review · Reviewer_XSZ1 · 2021-11-02

**Correctness:** 2
**Technical Novelty And Significance:** 1
**Empirical Novelty And Significance:** Not applicable
**Recommendation:** 5
**Confidence:** 4

**Main Review:**

**Strong Points:**
* In general, the paper is clearly written and easy to follow. The problem of depression detection is well-motivated, the overall presentation is good.
* The authors discussed adequately different related works to the problem of depression detection. They stated the gap of the existing approaches, which rely on handcraft features to detect depression.
* The authors describe well the preprocessing steps and create a balanced version of the dataset for training the model.

**Weak Points:**

While most of the paper is easy to understand, there are some points in the paper that lack clarity for the novelty and significance.

* The authors propose a flexible concatenation of different embeddings for a robust and rich tweet representation. However, the novelty of this method is inadequate compared with state-of-art contextualized embeddings such as BERT.  For example, the research work [1] addresses the same problem and considers contextualized embeddings (e.g., BERT, Roberta) as state-of-art-embeddings baselines. Here, I suggest the author benchmark their approach against BERT embeddings. For example, the authors can use BERTweet [2] as a baseline to SERCNN .
* In table 2, the authors show the hyperparameters optimized for the proposed approach and baselines. This is not clear to me that all models are optimized by the same hyperparameter values, given that these models have different architectures.
* Some details about the experiment's setup are missed. For example, how the authors split the data for training, validation, and testing the models (e.g., train-test split ratio). Is there overfitting during the model’s training?
* Further, stacking embeddings is also not memory efficient. I find the advantage of stacked embeddings is addressing out-of-vocabulary words. Here, I suggest the authors conduct more experiments to highlight this contribution in their experiments as well.
* Since the authors didn’t specify the test data used, It would be better to evaluate their approach on a different dataset (see [3]), which may contain out-vocabulary words, and can show the efficiency of the proposed approach.
* Also, I wonder if stacking embedding is efficient for this task only? Or It’s a generic, richer representation that can achieve good performances with very little data.
* In section 5, The authors analyzed the performance of their approach with little training data (e.g., 10, 30, 100, 500, etc.) I wonder how do you sample these tweets from the dataset, randomly?. Here, I recommend obtaining multiple samplings and performing a standard statistical test to benchmark how significant are the outperforming results.
* Although, this study lacks the explainability of depression detection, and the authors name it as future work. I find some studies [4] in 2020 that address depression detection with an explainable approach.


[1] Zhang, Yipeng, et al. "Monitoring depression trend on Twitter during the COVID-19 pandemic." arXiv preprint arXiv:2007.00228 (2020).
[2] https://github.com/VinAIResearch/BERTweet
[3] https://github.com/swcwang/depression-detection
[4] Zogan, Hamad, et al. "Explainable Depression Detection with Multi-Modalities Using a Hybrid Deep Learning Model on Social Media." arXiv preprint arXiv:2007.02847 (2020).





**Summary Of The Paper:**

This paper proposes a deep learning approach, called SERCNN, for depression detection from social media data (tweets). The proposed approach is flexible and stacks different embeddings (Glove pretrained model,  and a learned embedding vector from LSTM) for robust and richer tweets representation. Overall, the contributions of this approach are third folds: i) leveraging social media as a valuable source of information for depression detection; ii) the authors employ stacked-embeddings is a good technique to handle out-of-vocabulary words; and iii) Using a few training examples (e.g., 10 tweets per user), the proposed approach showed a remarkable performance (78% accuracy) in depression prediction.  Further, the authors conduct a set of experiments on a public Twitter dataset for depression detection and compare it with different baselines.

**Summary Of The Review:**

Overall, I rate this paper as marginally below the acceptance threshold (5). The problem is interesting, however, the proposed approach lacks a comprehensive evaluation with state-of-the embedding models to ensure the efficiency of the approach. For example, the authors can employ BERT embedding as a baseline to evaluate to which extent SERCNN approach is good. Further, I suggest the authors, clarify how do they perform sampling (e.g., 10 tweets) to assess their approach with little data.

---

### Official Review · Reviewer_pWBG · 2021-11-02

**Correctness:** 4
**Technical Novelty And Significance:** 2
**Empirical Novelty And Significance:** 1
**Recommendation:** 3
**Confidence:** 2

**Main Review:**

The motivation and the results of the paper are very interesting and definitely will be of importance in a more applied community focused on NLP or ML for mental health. However, the technical novelty for a venue such as ICLR is somewhat limited. All network components used exist already and while the different pre-training embedding models used appear to be effective, they do not teach us anything new about deep learning.

I would suggest to the authors to consider a different venue that appreciates this kind of findings.

**Summary Of The Paper:**

The paper proposes a concatenation approach to combine multiple social media texts through a stacked embedding layer and demonstrates its effect in depression prediction based on text.

**Summary Of The Review:**

(see above)

---

### Decision · Program_Chairs · 2022-01-20

**Decision:**

Reject

**Comment:**

This paper tackles a very important problem of detecting depression on Twitter. As the reviewers expressed in their reviews. this paper will be of interest for the community of researchers applying ML models to mental health domain. It is unfortunate that the authors did not respond to the reviewers' concerns and questions. I strongly encourage the authors to improve the paper based on the authors comments and questions and resubmit to a future venue.